# Mercury (Hg) Contaminated Sites in Kazakhstan: Review of Current Cases and Site Remediation Responses

**DOI:** 10.3390/ijerph17238936

**Published:** 2020-12-01

**Authors:** Mert Guney, Zhanel Akimzhanova, Aiganym Kumisbek, Kamila Beisova, Symbat Kismelyeva, Aliya Satayeva, Vassilis Inglezakis, Ferhat Karaca

**Affiliations:** 1The Environment & Resource Efficiency Cluster (EREC), Nazarbayev University, Nur-Sultan 010000, Kazakhstan; zhanel.akimzhanova@nu.edu.kz (Z.A.); aiganym.kumisbek@nu.edu.kz (A.K.); kamila.beisova@nu.edu.kz (K.B.); symbat.kismelyeva@nu.edu.kz (S.K.); aliya.satayeva@nu.edu.kz (A.S.); ferhat.karaca@nu.edu.kz (F.K.); 2Environmental Science & Technology Group (ESTg), Department of Civil and Environmental Engineering, Nazarbayev University, Nur-Sultan 010000, Kazakhstan; 3Environmental Science & Technology Group (ESTg), Department of Chemical and Materials Engineering, Nazarbayev University, Nur-Sultan 010000, Kazakhstan; 4Chemical and Process Engineering, University of Strathclyde, Glasgow G1 1XQ, UK; vasileios.inglezakis@strath.ac.uk

**Keywords:** acetaldehyde plant, chlor-alkali plant, contaminated sites, Lake Balkyldak, mercury removal, Nura River, site contamination, soil and sediment pollution, soil treatment

## Abstract

Mercury (Hg) emissions from anthropogenic sources pose a global problem. In Central Asia, Kazakhstan’s central and northern regions are among the most severely Hg-contaminated territories. This is due to two former acetaldehyde (in Temirtau) and chlor-alkali (in Pavlodar) plants, discharges from which during the second half of the 20th century were estimated over 2000 tons of elemental Hg. However, the exact quantities of Hg released through atmospheric emissions to the environment, controlled discharges to the nearby aquatic systems, leakages in the cell plant, and contaminated sludge are still unknown. The present review is the initiation of a comprehensive field investigation study on the current state of these contaminated sites. It aims to provide a critical review of published literature on Hg in soils, sediments, water, and biota of the impacted ecosystems (Nura and Irtysh rivers, and Lake Balkyldak and their surrounding areas). It furthermore compares these contamination episodes with selected similar international cases as well as reviews and recommends demercuration efforts. The findings indicate that the contamination around the acetaldehyde plant site was significant and mainly localized with the majority of Hg deposited in topsoils and riverbanks within 25 km from the discharge point. In the chlor-alkali plant site, Lake Balkyldak in North Kazakhstan is the most seriously contaminated receptor. The local population of both regions might still be exposed to Hg due to fish consumption illegally caught from local rivers and reservoirs. Since the present field data is limited mainly to investigations conducted before 2010 and given the persisting contamination and nature of Hg, a recent up-to-date environmental assessment for both sites is highly needed, particularly around formerly detected hotspots. Due to incomplete site remediation efforts, recommendations given by several researchers for the territories of the former chlor-alkali and acetaldehyde plant site include ex-situ soil washing, soil pulping with gravitational separation, ultrasound and transgenic algae for sediments, and electrokinetic recovery for the former and removal and/or confinement of contaminated silt deposits and soils for the latter. However, their efficiency first needs to be validated. Findings and lessons from these sites will be useful not only on the local scale but also are valuable resources for the assessment and management of similar contaminated sites around the globe.

## 1. Introduction

Mercury (Hg) and its common compounds (e.g., HgS (cinnabar), HgCl_2_) are persistent, highly bioaccumulating, very toxic to people and the environment [1]. They are released to ecosystems by both natural and anthropogenic processes [2,3]. Hg exists in the environment mainly in the form of elemental (Hg^0^), inorganic (i.e., ionic: Hg^+^ and Hg^2+^), and organic species (MeHg^+^, MeHg^2+^, EtHg^+^, etc.) [2,4]. Anthropogenic processes with the highest contribution to global Hg release include disposal of batteries [5] and energy-efficient lamps [6], mining and mine wastes [7], fossil fuel combustion (mainly coal) [8], leather tannery [9] and other industrial activities, including the manufacturing of chlor-alkali and caustic soda using Hg-cell. Moreover, due to the Hg cycle through the air, water, and soil, re-emissions of Hg from Hg-contaminated areas, including natural and anthropogenic sources of the past, are also significant drivers of the total Hg releases, especially to the atmosphere [10].

In addition to being persistent, Hg released to the atmosphere can travel long distances making localized discharges a global concern [1,11]. However, in general, atmospheric Hg concentrations are rarely above risk-inducing levels, and the major problem with long-range transport of Hg is its deposition and subsequent introduction to the food chain [1]. Hg in ambient air exists mainly in the form of elemental Hg [10,12,13], but there are more Hg compounds in water with varying solubilities: Hg(II) chloride is readily soluble, Hg(I) chloride and HgS are less soluble, and Hg^0^ is insoluble in water [14]. Inorganic Hg is then methylated in water by bacterial species (*Pseudomonas*) in biota, which leads to the formation of very toxic methylmercury (MeHg) that can enter the food chain of the aquatic ecosystem [14]. Waterbed sediments act as a buffer medium and, depending on the environmental conditions, can accumulate or liberate Hg [15]. Hg may be naturally present in soils in the form of Hg salts and minerals [16], and also come from the wet or dry deposition of airborne Hg or industrial discharges [13]. Atypical to compartments other than air, elemental Hg might be found massively in contaminated soils, for example, in the case of spillage [8]. However, more common forms of Hg present in soil include inorganic Hg (e.g., in HgCl_2_, HgO, HgS) and, given specific circumstances, organic Hg (not be mistaken with inorganic Hg salts and minerals attached to organic matter) [13]. Different forms of Hg in various compartments exhibit different physicochemical behaviors [16], which makes investigating the fate and transport of Hg extremely difficult [17].

There have been two major Hg contamination episodes in Kazakhstan: (1) a chlor-alkali plant operated in Pavlodar based on Hg-cell technology and (2) a carbide-aldehyde plant located on the banks of Nura River in Temirtau (Figure 1). The former PO “Khimprom” chlor-alkali plant (at present active as JSC “Pavlodar Chemical Plant” (PCP)) in Pavlodar operated in 1975–1993. It manufactured chemicals for both civilian and military uses; primarily, chlorine and caustic soda using Hg-cell method [18]. Several settling lagoons, including Lake Balkyldak (located north of the plant), were formed and explicitly used for industrial waste such as Hg-rich sludge [19]. The second, the Industrial Complex JSC “Carbide” (at present “Karaganda Plant SK Temirtau”) produced carbide and acetaldehyde. It has been the primary source of Hg contamination in Central Kazakhstan from 1950 to 1997 [20]. Although from the mid-1970s to the mid-1990s, the wastewater discharged from JSC Carbide has been treated, the discharges to the Nura River and Samarkand reservoir still had high Hg content. The sediments and river water in the vicinity of the main discharge drain still contain high Hg concentrations [21]. Discharges and losses from both of these facilities contained over 2000 t of Hg combined and created the major Hg contamination cases in Kazakhstan’s history, which are still not thoroughly assessed to this date.

The present paper is the initial output of a comprehensive site re-assessment project that aims to provide an overview of the current state of Hg contamination in Kazakhstan. Its objectives are (1) to review Hg-contaminated sites in Kazakhstan as situated in its northern and central regions, (2) to provide their detailed analysis and comparison with the literature, and (3) to review and summarize remediation recommendations for Hg-contaminated sites. We further aim the present review to serve as a resource at international level regarding the assessment, rehabilitation, and management of Hg-contaminated sites by similar industrial activities. Multiple comparable cases are present around the globe, particularly in Eastern Europe and East Asia where identical/related industrial processes using Hg have been employed.

Given the specifics of the contamination cases, the study’s scope was limited to media acting as a “sink,” i.e., soil, sediment, water, and biota. The scarce and fragmented literature on the subject has been critically reviewed. The current situation regarding the exposure assessment has been presented, and mitigation, remediation, and research recommendations have been addressed. A comprehensive literature review on Hg contamination in Pavlodar and Nura River regions has been performed using the platforms including international journal databases and local publications. Studies investigating the concentrations of several contaminants along with Hg levels have also been included. The majority of the studies reviewed in the present work have been published before 2010 as both contamination incidents occurred more than 20 years ago. Mainly, relevant high quality works published in the first (Q1) and second (Q2) quartile journals (as classified by Scimago Journal) have been targeted, with the following exceptions that have not been published in Q1/Q2 journals (determined case by case): (a) scientific works essential to discussion on Hg contamination, (b) related articles in Russian published in local scientific journals and/or university bulletins, (c) limited selected studies with high relevance yet published in third (Q3) quartile journals or below, and (d) pertinent legal documentation regardless of their publication date.

## 2. Human Exposure to Hg, Regulations on Hg, and Hg Mobility

### 2.1. Human Exposure to Mercury, Regulations

Hg is classified as one of 13 priority hazardous substances according to the adopted Water Framework Directive and the Environmental Quality Standards Directive [23,24]. Ingestion of contaminated products, water, vegetables grown on the polluted sites, inhalation, dermal contact, as well as pica-behavior of children (ingestion of non-edible objects such as soil) are the main exposure pathways [16,17]. The consumption of fish from Hg-contaminated sources is the main route of exposure to MeHg [25,26,27]. It is considered to be the most toxic form of Hg due to its increased retention by organisms at various levels of the food chain [28]. The sensitive groups, including the fetus, newborn, and children, are most susceptible to Hg exposure’s harmful effects on the nervous system, kidneys, and fetuses [2,16]. These effects are mainly dependent on the exposure pathway, duration, and concentration [29].

Different forms of Hg have varying human toxicity and bioavailability, strongly dependent on the route of exposure [18]. Elemental Hg is much more dangerous when inhaled (due to a high vapor pressure) than being ingested, as it has low gastrointestinal absorption, but it can easily penetrate lung tissue and enter the blood system [18]. On the other hand, inorganic Hg (e.g., HgCl_2_) and organic Hg (via food, mostly fish) are more harmful when ingested, and organic Hg has up to 95% bioavailability through the gastrointestinal tract [16,30]. Finally, the dermal absorption of all three forms of Hg is still unclear, and elemental Hg and HgCl_2_ have some potentials for dermal uptake [16]. Although dermal exposure to contaminated sites has generally received less attention than oral and inhalation exposures due to limited exposure scenarios and less perceived potential for toxicity, the risk can still be significant for specific contaminants and scenarios [31]. U.S. Environmental Protection Agency (U.S. EPA) introduced the Reference Concentration for Inhalation Exposure for elemental Hg (3 × 10^−4^ mg/m^3^). It is marked as non-classifiable as to human carcinogenicity (Class D) [32], whereas the Agency for Toxic Substances and Disease Registry [30] set the minimum risk level for chronic exposure to elemental Hg via inhalation at 0.0002 mg/m^3^. U.S. EPA has also established a Reference Dose for Oral Exposure (RfD) of 3 × 10^−4^ mg/kg-d for mercuric chloride (HgCl_2_) and identified it as a possible carcinogen (Class C) [33]. MeHg has RfD of 1 × 10^−4^ mg/kg-d (U.S. EPA 2001) and minimum risk level for chronic ingestion of 0.0003 mg/kg-d [30]. It has been identified as possibly carcinogenic to humans (Class C) [34]. The provisional tolerable daily intake of 5 μg/kg of body weight for total Hg with a maximum of 3.3 μg (MeHg)/kg of body weight as a general guideline was recommended [14]. Different bioavailabilities, toxicity levels, and effects of Hg forms make speciation of Hg essential during the risk assessment of Hg-contaminated sites.

To prevent human exposure to Hg and protect the population and the environment, governments and health protection agencies worldwide have established maximum allowable levels of Hg in different media (Table 1, including maximum permissible concentration (MPC) standards for Kazakhstan). The Kazakh standards regulating Hg levels in soils are comparable to others, but Hg limits for air and water are more stringent than those proposed by the World Health Organization (WHO), European Council (EC), and governments of the U.S. and Canada, setting an ambitious goal to achieve. Moreover, in some cases, e.g., to reduce atmospheric Hg concentrations, while countries like Kazakhstan set maximum allowable concentrations, others, including the U.S., Canada, and the EU region, opt for regulations controlling the Hg emissions instead.

### 2.2. Effect of Site-Specific Conditions on Hg Mobility

Hg is generally immobile in soils because of its extremely high affinity to organic matter and sulfur ligands [13]. Hence, elevated but immobile Hg concentrations are usually associated with soils with high organic content. However, Hg can be released into the atmosphere at high-temperature conditions (Figure 2) [47]. It was also suggested that mostly immobile Hg fraction bonds to coarse-grain-sized soil particles, which only becomes mobile during the flooding period [48]. Under anaerobic conditions, the microbial reduction of sulfates might take place in soils resulting in the formation of Hg sulfide (HgS, known as cinnabar), which is a chemically stable and highly insoluble form of Hg.

Hg adsorption also differs in soil types, and the highest sorption of low-mobility Hg is attributed to the finest size fraction, e.g., in clay, loams, and sands [13]. Hg sorption is also attributed to the elevated specific surface area and cation exchange capacity in clays [49]. Water-soluble and highly mobile Hg fraction is significantly correlated with total organic carbon content in the soil, assuming that total organic carbon binds the mobile forms of Hg [13].

As natural conditions on a contaminated site have a strong influence on the content and migration of chemicals in the environment, it is essential to be informed about site-specific parameters. The climate of Central and North Kazakhstan regions is classified as an extreme continental climate with strong winds, sharp seasonal changes between cold winters (average temperature of −15 °C in January), and warm summers (average temperature of +20 °C in July) [22,50,51]. Low annual precipitation (e.g., 250 mm) and much higher potential evaporation (e.g., 1000 mm) values indicate the semi-arid climate’s characteristics. Melting snow is the main water supply for most of the area’s rivers and surface water bodies, and it is typical to have annual spring floodings followed by subsequent drainage in summer [22]. Moreover, the difference between diurnal and nocturnal temperatures in the region might easily exceed 20 °C [51]. The vegetation consists mainly of grass with more vibrant plant growth along the riverbanks and planes flooded in spring.

According to soil maps developed by the Food and Agriculture Organization of the United Nations, most soils in Northeast Kazakhstan (including Pavlodar and the surroundings of Nura River) are classified as Kastanozems [52]. Kastanozems have relatively high humus and calcium ions content and are used primarily for agricultural purposes [53]. Hg’s high affinity to soils’ organic content impedes its mobility, and the spread and transport of Hg from contaminated soils to groundwater is expected to be low in the region’s Kastanozems. In addition, limited areas of the region and the Nura River’s riverbed soils are classified as Solonetz and Gleysols, respectively [52]. Moreover, low temperatures, as well as snow cover during long winters in the area, inhibit the volatilization of Hg and its escape to the atmosphere. However, dry and warm weather with strong winds in the steppes during the summer season might contribute to atmospheric re-emission and long-range transport of Hg. Finally, low annual precipitation prevents solubilization of soil-bound Hg except for annual flooding episodes. These episodes (in addition to increasing soil’s moisture content) might erode some portion of the soil and transport Hg containing soil particles. Based on the climate and environmental characteristics of North and Central Kazakhstan, Hg contamination in this region is expected to be primarily localized due to relatively low mobilization, solubilization, and volatilization of Hg.

The following sections review the Hg contamination in two sites in Kazakhstan: a chlor-alkali plant impacting the area around Lake Balkyldak and Pavlodar Region (Site #1), and an acetaldehyde plant impacting the area around Nura River and Temirtau Region (Site #2). A summary of the studies investigating these regions (Table 2), the comparison between these two sites (Table 3), and similar sites from the literature (Table 4) are provided in the tabular forms.

## 3. Site #1: Lake Balkyldak and Pavlodar Region

During the operation of the chlor-alkali plant in Pavlodar, Hg was used as a cathode in the electrolytic separation of chlorine and caustic soda from brine resulting in large quantities of Hg-rich sludge. A significant part of it has been discharged to the soil and atmosphere in the vicinity of the plant [19]. Some estimates of the Hg losses at PCP were of up to 1300 t [65], although some part of it was allegedly recycled and recovered, the rest (estimated as 700 t) is still unaccounted [50]. The approximate amount of Hg discarded to Lake Balkyldak has been estimated at around 135 t [63]. These releases affected different media in the region and significantly influenced the local population making the region an important research subject of numerous studies.

### 3.1. Soils

Hg concentrations at specific locations were extremely high, sometimes in the order of g/kg [63]. Several studies showed that the Hg concentration in soil exceeded the local permissible limit of 2.1 mg/kg, as well as international limits of 1, 6.6, and 0.83 mg/kg of the U.S. (California), Canada, and the EU (Netherlands), respectively. Mean soil Hg concentrations the most contaminated areas were reported to be 835.9 mg/kg [19], >1000 mg/kg [59], and almost 2000 mg/kg [63]. Average Hg in a broader area in the northern industrial zone of the city in 2001–2002 was reported to be 3.51 mg/kg [60], while mean Hg was 2.65 mg/kg in topsoil samples around Lake Balkyldak [50]. The limits for Hg in soils of industrial areas are higher than that in residential/agricultural soils (Table 1), these reported average concentrations did not exceed the regulations. Two soil sampling campaigns run in Pavlodar and nearby Pavlodarskoye village in 2001–2002 reported mean Hg lower than 2.1 mg/kg and 1.8 mg/kg in the city [60]; and 1.04 and 1.5 mg/kg in the village [19,57]. Panin and Geldymamedova [60] reported that mean concentrations of several potentially toxic elements (PTEs) (namely Hg, Cd, Co, and Mo) were 1.6-22.5 times higher than their background values. However, the sorption of Hg is not much related to the presence of any other PTEs [13]. Overall, excessively high concentrations have been detected only on the territory of PCP, but not in the inhabited localities.

### 3.2. Sediments

Hg contamination is present in the sediments of Lake Balkyldak and is more profound in the vicinity of the wastewater outfall pipe (within 1000 m) with the mean and peak concentrations of 167 mg/kg and close to 1500 mg/kg in 2001–2002, respectively. The Hg content of sediments from the old river channel was less than 0.050 mg/kg [50]. Occasionally elevated Hg concentrations (40–60 mg/kg) were detected in the same period in more distant locations of the lake (more than 5 km from the outfall), possibly linked to sediment transport or disposal of Hg-rich waste at other places. Besides, similar to Nura River sediment–water interaction, resuspension of the sediments (especially from the top layer) might be one of the most critical water contamination sources with Hg [50]. Finally, highly toxic forms of Hg (MeHg and other organic Hg compounds) were found in sediments and fish, possibly linked to bioaccumulation in aquatic food chain [65]. These values may now be deemed outdated, and further studies on Hg’s distribution and speciation in the lake’s sediments are recommended.

### 3.3. Water

Lake Balkyldak, a wastewater storage pond, has contamination localized, particularly near the discharge pipe in the southern part of the lake. According to Ullrich et al. [50], Hg concentrations in the lake water were 0.11–1.39 μg/L in 2001–2002, increasing to 7.3 μg/L on windy days due to disturbed sediments. After the Hg pollution containment project in 2005-2011 was completed, it was reported that concentrations of Hg in the surface waters of the lake decreased from a maximum of 3000 ng/L in 2001 to 300 ng/L in 2008, since the major part of Hg waste was sunk [63]. The latter value did not exceed the local regulatory Hg concentrations, WHO, the U.S. and Canada (Table 1), but was higher than 0.07 μg/L, maximum contaminant level in surface water by the EU regulation [39].

The analysis of water in the Irtysh River and oxbow lakes in 2001–2005 showed traces or undetectable levels of Hg [43,50,59]. A thorough assessment of the regions’ groundwater, including Hg measurements and modeling, concluded that the movement of the Hg plume is headed to the north in parallel with the river, which implies that the river is not currently receiving polluted groundwater [59,63]. Moreover, groundwater in Pavlodarskoye village is used for local consumption, and its Hg levels did not exceed the level of detection (5 ng/L) in 2001–2002 [57]. In 2005, a plume of contaminated groundwater (up to 150 μg/L) was revealed to spread in the plant’s north-northwestern direction [59]. Six years later, when the containment project was finished, Ilyushchenko et al. [63] reported that the plume continued to spread. Using computer models, the following conclusions were made (assuming the hydro-geological conditions remain unchanged): (i) groundwater contamination in the east direction is expected to decrease, and (ii) Hg levels in the river water would be in the safe range. Overall, it is expected that the contamination will not significantly affect water in the Irtysh river; however, the Hg levels in lake Balkyldak might be higher. In order to confirm the statement, performing more recent measurements of Hg levels in groundwater are highly recommended.

### 3.4. Biota

Fish consumption is considered to be the primary source of exposure to MeHg; thus, the region’s inhabitants who catch and consume fish from the lake and river could be under certain risks associated with Hg exposure. While Kazakhstan’s limit on maximum permissible Hg concentration in non-predatory fish is 0.3 mg/kg, the detected levels in fish from the lake were 0.18–2.2 mg/kg, in 50 out of 55 samples exceeding the defined limit [58,59]. Comparing Hg in perch caught from Lake Balkyldak and the Irtysh River, it was obvious that contamination in the lake has been a serious issue: the mean Hg concentration in samples from Lake Balkyldak was 0.89 mg/kg (range: 0.16–2.20 mg/kg), while the samples from Irtysh River had a mean concentration of 0.112 mg/kg (range: 0.075–0.125 mg/kg) [57]. This finding was in line with the results reported by Ilyushchenko et al. [59]: Hg range in predatory fish from Irtysh River and oxbow lakes was 0.075–0.16 mg/kg. Ilyushchenko et al. [63] reported that Hg concentrations in the lake biota generally decreased, but Hg levels in some fish in 2007 were 1–1.5 mg/kg, which were at least twice higher than regulatory limits (Table 1). However, it is necessary to note the limited sample size in these studies and the fact that they were conducted before 2007. In terms of flora, several plants in the area 1-1.5 km away from PCP were found to contain 1.09–1.66 mg/kg of Hg in the aerial parts. However, Hg generally tends to accumulate in the root without traveling to the higher parts of the plant [57].

### 3.5. Air and Snow

Hg can be present in several inter-converting species in different media in the environment, including air, soil, water, sediments, and biota. Since the release took place several decades ago, concentrations of Hg vapors are expected to be very low unless the Hg release from contaminated soils, sediments, and water still occur on the site. According to the measurements in 2006, Hg vapor concentrations in the ground-level air were in the range of 100–1600 ng/m^3^ (MPC of 300 ng/m^3^ exceeded in 7 out of 16 tested points) [63]. They also detected a Hg vapor concentration of >10,000 ng/m^3^ at one sampling point on the industrial site.

Besides, snow is considered an ideal matrix for observing atmospheric precipitation because it accumulates a great amount of the contaminant due to a larger surface area of snowflakes. Shakhova et al. [64] evaluated the Hg content in snow and found that the Hg content in solid snow covers in the vicinity (0.5–2.5 km) of the chemical plant’s site varied greatly and exceeded the background values of 0.15 mg/kg by 1.5 to 7 times. In NE zone, the reported Hg content was 0.31–1.04 mg/kg, in SW—0.22 mg/kg, and in NW—0.03–0.26 mg/kg; in Pavlodarskoye village, the value was close to the background. Elevated Hg concentrations in the NE zone might be related to Hg contamination from past industrial activities and wind directions. The results showed that air pollution is a real concern in the region; thus, monitoring atmospheric Hg and its control program are strongly recommended.

### 3.6. Hg in Population and Food

Shaimardanova et al. [62] measured concentrations of Hg and other PTEs in hair samples collected from the children in Pavlodar. They suggested that high hair Hg content (0.44 ± 0.5 mg/kg) was associated with the activities of high-ash coal thermal power, metal processing plants, and the chemical industry. The most significant Hg accumulation (0.5–0.7 mg/kg) was detected in the children’s hair living in western, southwestern, and northwestern districts of the city close to the industrial zone. Moreover, some inhabitants of Pavlodarskoye village own livestock, and as a result, drinking Hg-contaminated surface water might pose a risk to cows grazing on the territory around PCP. Hence, there is a significant risk of Hg exposure via milk and beef consumption. Ullrich et al. [57] assessed the Hg concentrations in bovine milk and tissue samples and reported the values to be < 2 μg/kg and 10.96 μg/kg, respectively. Thus, the Hg contamination in the industrial zone’s vicinity is unlikely to pose health risks to the livestock. However, further studies might be necessary to determine if there is a higher risk for people who consume large quantities of beef liver and kidney as these are the organs with a significant Hg bioaccumulation [93].

### 3.7. Human Health Risk Assessment by Hg Exposure

Since the significant fraction of Hg accumulated in fish is present in the organic form, specifically MeHg, consumption of fish from Lake Balkyldak is a crucial issue that was investigated in 2001–2002 [57]. According to the THg values measured in 55 fish samples, 50 of them contained >0.3 mg/kg MeHg (assuming 100% of Hg is MeHg). The analysis of fish samples caught from Lake Balkyldak in 2006–2007 showed that high Hg levels (1–1.5 mg/kg) were still present in the pond, hence, posing health risks to the local population [63].

Moreover, Woodruff and Dack [19] suggested possible health risks to a local population at village Pavlodarskoye caused by homegrown vegetable consumption due to the potential ingestion of the contaminated soil attached to the vegetables. They used two different risk assessment models from the UK and Netherlands that are mostly based on the intake rate of selected vegetables (with varying rates of accumulation and preparation methods) by the local population. Contaminated Land Exposure Assessment model was used for the case of ingestion of vegetables and soil attached to them. Van Hall Institute Risc-Human Model Version 3.0 for Risk Assessment for Soil Contamination was used for cases of soil ingestion by children, and consumption of meat and vegetables by both children and adults. The estimated risks for children were higher than for adults in all exposure scenarios, possibly due to lower body weight. Another finding was that hazard quotients (HQ) calculated for the scenario of contaminated meat ingestion were higher than the contaminated vegetable intake; however, it is necessary to keep in mind that the dietary patterns vary among people. Besides, the ingestion of contaminated soil poses a high risk due to increased concentrations despite small doses. HQs were in the range of 3.05–3.06 for consuming contaminated vegetables by female children and 6.67 for the ingestion of soil attached to vegetables by female children. Although the authors used these methods, they refrained from clear conclusions due to several limitations that may have affected the risk assessment. For instance, they did not take into consideration Hg bioaccumulation and difference in vegetable uptake; amounts of ingested soils; and absence of measurements of Hg levels in hair, blood, and urine.

## 4. Site #2: Nura River and Temirtau Region

The main river of the Central Kazakhstan region is the Nura River, which has a total length of 978 km and a catchment area of 60,800 km^2^ (Figure 1). All rivers of the Nura-Sarysu basin are mainly of snow nutrition; therefore, the bulk of the annual runoff occurs during the flood period in spring. The Samarkand reservoir is situated on the Nura River 9 km downstream from Temirtau and has a total storage capacity of 254 million m^3^ [22].

The production of acetaldehyde at JSC Carbide was performed using the Kucherov method requiring Hg(II) sulfate salt as a catalyst, while the contact acid was regenerated with the use of Hg [20]. Although from the mid-1970s to the mid-1990s, the wastewater discharged from JSC Carbide was treated using various techniques including neutralization, sulfidation, magnetic treatment, the discharges to Nura River (2.5 km from Temirtau) had extremely high Hg content possibly due to inefficiency of the treatment (Table 2). Moreover, a shallow Intumak reservoir located 75 km downstream of Temirtau serves as a settling pond for soil particles contaminated with Hg.

### 4.1. Soils

Several studies show that Hg discharged from the acetaldehyde production plant contaminated different media, including the river floodplain’s soils. Thus, Heaven et al. [55] estimated that topsoils of the floodplain along 75-km-long part of the Nura River (from Samarkand reservoir to Intumak reservoir) contained approximately 53 t of Hg, and the concentration range spanned from near background levels (0.01 mg/kg) to more than 100 mg/kg. Moreover, it was found that the majority of the contaminated materials such as topsoils and technogenic silts (70–90%) were located in the first 25 km from Samarkand reservoir [66]. Moreover, the mean concentrations of Hg in soil were 5.9 mg/kg (12.5 km downstream) and exceeded the Kazakhstani standard of 2.1 mg/kg (Table 1) up to ten times and the limits of the U.S. (California) of 1 mg/kg and of EU (Netherlands) of 0.83 mg/kg in this region [55]. However, Hg concentrations in topsoils reduced dramatically with distance from Temirtau, but it still reached the value of 10 mg/kg at 60 km downstream. Besides, several local hotspots with high Hg content were reported by Ilyushchenko et al. [66]: Swamp Zhaur (62 t), old ash lagoon of KarGRES-1 (32 t), wastewater treatment facilities, and banks of the main drain from acetaldehyde production plant (10 t). In Zhaur swamp, which was formerly used as a waste disposal area, mean and maximum concentrations of Hg in soils were 307 mg/kg and 1974 mg/kg, respectively.

Moreover, Hg concentrations in soils also tended to be the highest on surface layers and decreased with depth [55]. Although the mobility of Hg in soils is generally low, it is more mobile in silts and sediments and, as a result, can potentially be mobilized to water. The estimated amount of Hg accumulated in the bank silt deposits near the riverbanks was 65 t with a mean concentration of 73.3 mg/kg in the most contaminated section falling to a mean of 13.4 mg/kg at 70 km downstream of the river [55].

### 4.2. Sediments

The most polluted sediments were found within 15–20 km downstream of the plant’s discharge point with their generally limited transport except during annual flood episodes. Different studies reported the estimated total amounts of technogenic silts formed from the power station fly ash in the riverbed between Samarkand and Intumak reservoirs to be about 463,500–550,000 m^3^ and contain approximately 10 t of Hg [54,66]. Besides, the total amount of Hg emissions from the acetaldehyde production plant were estimated to be 1200 t [67]. A wastewater pipe from the plant discharged wastewater directly to the Nura River and is considered one of the most contaminated parts close to the city. The measured Hg concentrations in the range of 150–240 mg/kg in sediments are the highest within the first 15 km [54], and it was 9.95–306 mg/kg in the most polluted section according to another study conducted in 2001–2002 [61]. Agreeing to the Hg concentrations in soils, technogenic silts, and river sediments, the first 10 to 20 km from the wastewater discharge point were confirmed to be the most contaminated part of the river [54,56,61,66]. Ullrich et al. [61] reported that the highest MeHg concentrations in the surface sediments were ranging from 4.9 to 39 μg/kg, but usually less than 0.1% of total Hg (THg). A significant negative correlation was found between total Hg concentrations and the MeHg fraction in the sediments.

Regarding the suspended particulate matter, Ullrich et al. [61] reported that Hg in surface waters was mainly adsorbed to particulates. Many international studies reported a significant positive relationship between total Hg and total suspended solids downstream of contaminated sites [48]. However, it was in strong contrast to an earlier study on the Nura River, where the majority of Hg downstream of the outfall was found to be in the dissolved form [54].

Overall, Hg quantities in all media tend to fall with distance along the river, because a significant part of polluted sediments is not moved far downstream except during springtime floods [54,55,61].

### 4.3. Water

Riverbanks erosion and floods may disturb contaminated material such as sediments, silts, and soils, which might result in remobilization and release of Hg to water [61]. As an example, an increased flow during the flood in 1997 resulted in relative growth of THg by 235% (Hg concentrations increased from approximately 0.5 μg/L to 1.25 μg/L) in the river water near the city of Temirtau [54]. Moreover, during the flood in 2004, Hg concentrations in surface waters amounted to 1.6–4.3 μg/L [61]. In contrast, during other seasons, Hg concentrations in the river usually did not exceed 0.5–1 μg/L but in several cases were higher than 0.3 μg/L [54,66]. According to Table 1, Kazakhstani regulations for Hg concentrations in fish-inhabited waters are more stringent than other international limits, so the values might not comply with several regulations. It is also necessary to consider different limits of detection of the used equipment, sampling locations, time of the year, and other important factors.

According to Heaven et al. [54], the estimated total amount of silt deposited in backwaters was approximately 161,500 m^3^, and the estimated Hg quantity was 3.5 t. It was suggested that the prevailing part of the silt and Hg in backwaters were located in the first 25 km section of the river and that Hg content in backwaters was directly related to Hg in riverbed silts.

Talking about Hg in the Intumak reservoir 75 km downstream of the river and other terminal wetlands in the Korgalzhyn National Park, the results of different studies showed that the concentrations were higher than the background (<5 ng/L) [54,61]. However, it is recommended to collect and analyze more recent samples, since more than ten years have passed since the last analyses.

### 4.4. Biota

Consumption of local home-produced food might result in human bioaccumulation of Hg, and more importantly, consumption of contaminated fish and shellfish is considered the main MeHg exposure pathway, since cooking does not remove Hg from fish. Some people living in Temirtau and surrounding villages consume home-grown vegetables, beef, milk from cows and other livestock, and fish caught from the river [3]. Several studies confirm the bioaccumulation of Hg in the aqueous food chain. Thus, Ullrich et al. [61] showed that Hg concentration in biota in the most polluted section of the river was 15–20 times higher than the background level, and THg contents in water or sediments cannot be used for prediction of Hg in fish tissue. Ilyushchenko et al. [66] reported that Hg in fish and aquatic plants did not exceed the locally defined limits of 0.6 and 0.3 mg/kg for predatory and non-predatory fish (also complied with international limits from Table 1). The more recent study of fish samples caught locally or purchased from Temirtau markets showed that more than 33% of the samples contained Hg > 0.5 μg/g. Approximately 84% of the samples had Hg > 0.3 μg/g, which indicates the possibility of bioaccumulation of Hg in the aquatic food chain [3]. On the contrary, the results from a study conducted in 2005 reported that crayfish bought on the local market in Temirtau contained 0.026 mg/kg of Hg, while crayfish caught from the reservoir contained about 0.043 mg/kg [61]. According to the same study, Hg was found in fish samples more than 125 km downstream, indicating a significant transport of dissolved MeHg to further regions and/or MeHg entering the biological chain in situ in those downstream areas.

Talking about aquatic plants, Hg concentrations in narrow-leaf cattail (*Typha angustifolia*) from Nura Riverbanks were in the range of 0.03–0.63 mg/kg with the maximum being near the effluent outfall canal [61]. Although Hg concentrations in technogenic silts were found to be higher than in the epiphytes from the same locations, epiphyte suspension still can be used as an indicator to show the extent of pollution in the watercourse [56].

### 4.5. Hg in Population

Hsiao et al. [3] studied the relationship between human exposure to Hg and its concentration in the Temirtau region’s citizens’ and nearby villagers’ hair samples. Hg in their hair ranged from 0.009 to 5.184 μg/g with an average value of 0.577 μg/g, which was twice as much as in the control group from the Almaty region (considered as non-polluted by Hg). Nearly 17% of the tested population exceeded 1 μg/g for hair Hg, which corresponds to the reference dose (RfD) of 0.1 μg/kg body weight/day developed by the U.S. EPA [34]. A positive correlation was found between Hg concentration in hair and the consumption frequency of locally caught fish as well as the age: Hg content in hair of people over 45 years old was higher on average.

### 4.6. Comparison Between Cases of Pavlodar and Nura

In the Hg contamination cases of Lake Balkyldak and Nura River (Table 3), the main difference is that in the former, the lake was used as an independent wastewater lagoon and received approximately 135 t of Hg [63], whereas Nura River received treated water as well as accidental Hg releases in a total amount of around 1200 t [20,67]. The literature review showed that Hg concentrations in soils of the Pavlodar site’s most contaminated areas (see Section 3.1) were 3–6 times smaller than Hg concentrations at the local hotspots at Swamp Zhaur located near Nura River (see Section 4.1). On the other hand, the Hg concentrations in water remained below the limits in both cases (see Section 3.3 and Section 4.3) with the sudden increase in Hg concentration during annual floods and windy days, probably, due to a resuspension of the contaminated sediments [50,61]. It was also found that the Hg concentrations in sediments are the highest at wastewater outfall pipes (see Section 3.2 and Section 4.2) and sediment Hg levels of the most polluted parts are approximately equal—160 mg/kg in the lake and 150–240 mg/kg in the river [50,55]. The reason is related to the increase in specific surface area of the sediment particles thus Hg is accumulated more in the specific geologic substrate [7]. Measurements of Hg concentration in plants, fish, and the local population’s hair showed that Hg levels of those in the region of Lake Balkyldak are at least two times higher than those in the Nura River region (see Section 3.6 and Section 4.4). Overall, the Hg contamination of the Nura River might affect a larger area because of contaminant transport, but for the same reason, the river is expected to recover faster naturally compared to Lake Balkyldak which is a closed reservoir.

## 5. Comparison with Cases from Literature

Table 4 presents some of the most studied Hg contamination cases globally due to either acetaldehyde or chlor-alkali production plants, including Kazakhstan cases. Industrial facilities in all of the presented cases either have reduced their Hg emissions significantly, switched to Hg-free production technologies, or have been decommissioned due to the impact of Hg released to the local environment for often decades. Both Hg contamination cases in the Nura River and Lake Balkyldak seem to be more severe than the reported global cases listed in Table 4. The reviewed cases not only have higher estimated Hg discharges to the environment (1200 and 1000 t, respectively) but also exhibit generally higher Hg levels in soils, sediments, and surface waters in the vicinity of the facilities.

Several studies have investigated the extent of Hg contamination originating from chlor-alkali plants utilizing Hg-cell technology and its impact on the surrounding environment’s soils, sediments, and water. Operation of the chlor-alkali plant located in Flix, Spain, with a production capacity similar to the one in Pavlodar (around 100,000 t of Cl_2_/y) caused one of the major episodes of Hg pollution of the river bed materials. A delta-shaped sludge deposit containing approximately 10–18 Mg of Hg [83] had been accumulating for several decades at the base of one of several dams along the river in the absence of natural dilution and burial with river sediment material [81,82]. As a result, Hg concentrations as high as 640 mg/kg were measured in the accumulated deposits [82]. These are close to the range (0.11–617 mg/kg) reported by Ullrich et al. [50] for Lake Balkyldak sediments. Fluctuations of Hg concentrations of river sediment cores sampled in the vicinity of a chlor-alkali plant in Rm Valcea, Romania indicate Hg level spikes during floods, which can lead to transport of the site’s contaminated soils to the river [91], which might also be the case during annual flooding in Kazakhstan. Local authorities of Flix, Spain, opted for dredging of the polluted deposits [81,82]. However, Bravo et al. [91] strongly discouraged any treatment actions in Rm Valcea involving rework or dredging of the sediments to avoid resuspension of highly contaminated particles, which are buried (around 90 cm and deeper) in contrast to the ones in Flix. 

Another Hg pollution episode occurred in Estarreja, Portugal, with 8 km2 around a chlor-alkali plant marked as an extremely contaminated zone (>1.5 mg/kg of soil) with most of the soil Hg bound to silts and clays (0.063 mm), but as in the case of Pavlodar, the risk of Hg leaching to groundwater is still present [84]. Several studies on Hg contamination from chlor-alkali plants comparing their results to other similar works highlighted relatively elevated Hg levels in water [85] and soils [85,94] of the Lake Balkyldak region. However, the Hg content of the lake’s sediments was one of the highest globally [82,85,89,91,94]. 

Amidst all global cases of Hg contamination originating from acetaldehyde plant operations, Hg pollution of Minamata Bay is arguably the most impactful: population of areas surrounding Minamata bay was exposed to Hg mainly through the consumption of fish, leading to over 2200 registered cases of Minamata disease (Hg poisoning) in the area by 2003 [72]. The dredging project was conducted in 1974–1990 to decrease the Hg levels in sediments below 25 mg/kg by removing contaminated sediments and soil washing. As a result, the Hg level was reduced from 2000 to less than 10 mg/kg [71,72], which is considerably lower than the sediment Hg level in the Nura River (9.95–306 mg/kg, [61]). These measures have also led to a decrease of Hg content of water (total Hg of 1.3–4.3 ng/L, [71]), which are 1000 times lower than peak Hg concentrations in water of the Nura River during floods [61]. 

Two other aldehyde production sites polluted with Hg are Schkopau, Germany, and Guizhou, China. Both of these sites exhibit extremely elevated Hg level in the soil close to the plant (over 1000 mg/kg in Germany [51] and around 300 mg/kg in China [74,75,76], which are relatively lower than the maximum value reported for the Nura River (1974 mg/kg [66]). Out of all presented cases, the contaminated site in China is the most comparable to the one reviewed in this paper. The acetaldehyde plant also discharged Hg-containing wastewater into the river, and its polluted water (total Hg up to 1830 ng/L [75] was used for agricultural purposes, namely for irrigation of rice fields, leading to human exposure to Hg in both rice and fish from the river. Despite a large disparity in estimated Hg releases of these two facilities, both river systems had similar Hg concentrations in water.

Hg levels in some media show substantial reduction compared to previously reported values such as in the cases of the contaminated river near a Hg-cell chlor-alkali plant post-closure in New Brunswick, Canada [87] and Minamata Bay (Japan) close to an acetaldehyde plant after a massive dredging project [69,70,71]. However, all of the studies mentioned in Table 4 report concerns associated with one or several of the following: wet and dry deposition of Hg in soils, a potential re-release of Hg from sinks including soils and sediments, leaching from wastes disposed on the plants’ territory, and bioaccumulation of Hg in the site’s biota. There are increasingly more reported cases of Hg pollution resulting from operations of chlor-alkali (e.g., Sagua La Grande, Cuba [95,96]; Angara River, Russia [97]; Neratovice, Czech Republic [98]) and aldehyde plants (e.g., Ravenna, Italy [99]) worldwide indicating how prevailing is the problem. They also demonstrate the value in reporting the current situation and sharing lessons learned in environmental assessment, risk characterization, and remediation responses of all individual cases.

## 6. Remediation Responses

Elemental inorganic contaminants such as As, Cd, Cr, Cu, Hg, Mn, Ni, Pb, Ti, and Zn, are not degradable in the environment, and therefore, their remediation generally involves their removal or immobilization (Table 5). Several programs of demercuration of the contaminated sites in Kazakhstan were launched, but their completion level is uncertain, and their effectiveness was not adequately evaluated. Both the literature and the present review address certain further remediation alternatives; however, it should be noted that these should be validated based on the latest field data and pilot-scale laboratory tests before their implementation. Table 5 and Figure 3 present remediation technologies for different media including but not limited to those applicable to the Hg-contaminated zones in Kazakhstan.

A wide variety of Hg remediation technologies are available for different media (soil, water, air, waste), which could be categorized into main groups in terms of removal mechanism: physical, physio-chemical, chemical, and biological (Figure 3) [100]. Some of the techniques are established and thus have been used on contaminated sites for a long time, whereas others are still in the stage of development and testing. As each of these methods has its own advantages and limitations, an appropriate selection is only possible after a thorough investigation of the Hg-contaminated site since a lot of factors must be considered. These include the type of contaminated media, Hg speciation and concentrations, soil properties, environmental conditions, budget, and proximity to settlements.

Regarding emerging technologies, recent reviews point out nanotechnologies and the use of synthetic materials for Hg adsorption—innovative materials developed to enhance selected parameters including surface area and porosity [101]. This technology, depending on the material selection, can be applied for heavy metals removal from water, soil, and air; many studies reporting high adsorption values [101,102]. Another promising removal method seems to be phytoremediation of Hg from contaminated soils (via for instance, Brassica juncea (Indian mustard), Jatropha curcas, Polypogon monspeliensis (beard grass) [100]). It is found to be a safe, inexpensive, ecologically safe, and efficient demercuration method [103].

In order to increase the removal value and cost efficiency of remediation techniques (Table 5), inventions have been proposed. For example, while conventional microbiological treatment involves several steps aimed to reduce Hg(II) to Hg(0), a new proposed method (“Microbiological Mercury Removal from Contaminated Materials”) [104] offers a single-step treatment by a culture of microorganisms belonging to the genus Bacillus. This allows the removal of a broad group of Hg compounds (organic and inorganic) while omitting the pretreatment stage of leaching the Hg with chemical compounds. Moreover, Lestan et al. [105] invented a batch process, compared to conventional soil washing, providing advantages in the removal of alkaline precipitation of toxic metals hydroxides including Hg due to the alkaline adsorption of polysaccharides adsorbents. This eliminates the necessity of further cleansing of process waters; and in addition, allows the reuse of Ca-containing base and polysaccharides adsorbent which significantly decreases the generation of waste materials. Another technique introduced by Alden et al. [106], which can be implemented in situ and applicable for industrial pollution cases, (“Method and a Chemical Composition for Accelerated In-Situ Biochemical Remediation”) proposes the acceleration of the subsoil matter reduction by addition of a mixture comprising ferrous sulfide followed by the addition of organic hydrogen donor, which creates anaerobic conditions for bacteria to biodegrade residual contaminants. A final example regarding stabilization/solidification remediation technologies is a novel method (“Stabilisation curing agent and administering method that heavy-metal contaminated soil or solid waste are administered”) that uses a mix of magnesia, sulfur-based, and silicon-based compounds to provide long-term stabilization effect to various heavy metals compounds including Hg [107]. The proposed remediation technology, in comparison to the conventional method, minimizes the probability of solidified agents to re-dissolve within the contaminated matrix.

### 6.1. Demercuration of Lake Balkyldak

The U.S. EPA financially supported the demercuration plan of Lake Balkyldak and PCP territory and initially included the following: (1) dismantling the electrolysis factory, (2) constructing an underground landfill for disposing of uncontaminated demolished construction waste and other hazardous wastes of Class IV (flammable solids, spontaneously combustible substances and water-reactive), (3) excavation and soil washing, (4) constructing a “cut-off wall” around the territory of PCP to prevent further spread of Hg, and (5) excavating the concrete floors of the former facility followed by its thermal treatment [59]. However, due to limited investment, only the following steps were completed: (1) “cut-off wall” of total length 3588 m constructed; (2) the contaminated topsoils were excavated and isolated with the cut-off walls; (3) the hotspots (over 180,000 m^2^) were covered with clay; (4) the shop for Hg electrolysis was dismantled and disposed of in a special cell, then stabilized with cement and covered with asphalt layer; (5) a monolithic storage facility was constructed. According to the report on the development programs of Pavlodar city for 2016–2020 [124], the design specifications and estimates for the construction of an anti-filtration dam from the west side of the former pumping station to Balkyldak reservoir were developed for further implementation. Because the demercuration measures were limited to confinement actions, large quantities of Hg remained on the territory, dispersed in various media: soil, groundwater, and the lake’s surface water and sediments; thus, further demercuration is strongly recommended [63].

The media with the highest Hg content are soils of the plant’s territory and sediments of Lake Balkyldak near the outfall; therefore, the recommended remediation methods are directed primarily to these areas. Moreover, since the contamination was generally considered as localized, cost-effective containment strategies might be preferred over more complex removal methods. However, initially planned ex situ soil washing and thermal treatment of the concrete waste have not yet been performed. After completing the post-containment management and monitoring of Hg pollution on the PCP site in 2005–2009, Ilyushchenko et al. [63] suggested soil pulping and gravitational separation of elemental Hg as an option for soil remediation. As an alternative to containment, Ullrich et al. [50] suggested applying the combination of ultrasound and transgenic algae to treat the sediments, and according to preliminary trial results, ultrasound treatment achieved >95% efficiency in Hg removal. Electrokinetics (Table 5) might also be suggested as an alternative soil remediation technology (currently at the testing stage) for inorganic Hg species since it can be applied to all soil types, does not require excavation, can remove other contaminants, and is relatively inexpensive [112].

### 6.2. Demercuration of Nura River

Starting from the 1950s, the acetaldehyde factory in Temirtau discharged untreated wastewater containing Hg directly to Nura River, and only beginning in 1977, Hg-rich sludge had been allegedly neutralized, sulfidized, and magnetically treated before being discharged. This treatment system for contaminated wastewater was based on sulfide precipitation and reduced only metallic and some fraction of ionic Hg, but not organic Hg. In 1980, the aldehyde facility was under reconstruction, and the wastewater treatment system was dismantled. Clean-up of the Nura River Project is a joint project subsidized by the Government of Kazakhstan and the International Bank for Reconstruction and Development was conducted in 2004–2011 with the total project cost initially estimated at 67.82 million USD but increased to 97.42 million USD [125]. The results were rated as «satisfactory»: “Carbide” facility was demolished, all metallic Hg was immobilized and/or collected; the factory’s foundation and contaminated soil were landfilled. The cleanup project’s goal was to decrease Hg content in the river water downstream of the Intumak reservoir (Figure 1), and the target value of 50–120 ng/L in surface waters was achieved. A hazardous waste landfill for disposing of Hg-contaminated soil and sediment was constructed in 2010: the landfill received more than 1 million m^3^ of contaminated material [125]. However, according to Dushkina et al. [126], in 2014, the total Hg emissions to Kazakhstan’s environment were estimated to be 577,000 kg. The quality monitoring of the Nura River surface waters in 2017 showed that the maximum permissible concentration for Hg of 10 ng/L was exceeded in 17 out of 25 samples with a peak concentration of 4800 ng/L [126].

As mentioned in Section 4, the contaminant in the river is naturally diluted and may recover from the contamination faster if the Hg discharge ceases. One of the critical sections in terms of Hg-contaminated soil and sediments is the first 10–20 km from the wastewater discharge point because it also affects Nura River’s water. There is a lack of the reported Hg concentrations from the last ten years, so before developing and implementing a more effective demercuration project, a detailed analysis of the region’s soil, sediments and water is highly recommended.

Removing contaminated silt deposits and topsoils from the riverbanks and controlling wastewater discharge are highly recommended actions that can be taken to demercurate the region [54,55,66]. It was also recommended to rebuild a reservoir as a settling basin on Intumak as planned initially and reduce the spring floods that cause sediment disturbance [66]. Other contamination regulation actions may include Hg isolation by stockpiling silts under a thick layer (1 m) of inert cover material in a location safe from groundwater intrusion and flooding, removing and isolating the upper 40 cm of highly contaminated soil from Zhaur Swamp, and prohibiting the use of soils with Hg higher than 10 mg/kg for agricultural purposes.

## 7. Conclusions and Recommendations

The current paper presented a detailed review of Hg-contaminated sites (two former industrial sites) in Kazakhstan, including a comparison with the literature and a review of remediation alternatives. It aims not only to provide a critical update regarding the situation with these sites but also to serve as a resource for similar global Hg-contaminated sites that might assist in their assessment, rehabilitation, and management. Regarding the first site (Lake Balkyldak site), it can be concluded that the chlor-alkali plant, which stopped operating more than 30 years ago after having discharged hundreds of tons of mercury (Hg) to the environment, seriously affected the region and negatively impacted the health of the people from Pavlodar region and nearby villages.

Sediments from Lake Balkyldak are severely contaminated in contrast to the nearby Irtysh River, which was less affected by Hg pollution.Several hotspots with high Hg levels, mainly located on the chemical plant’s territory, might still be significant sources of pollution, and as a result, Hg from soil can affect the atmosphere, groundwater, and other media.Human health risks are mainly related to fish consumption from the lake and homegrown vegetables with the possibility of soil ingestion.

Regarding the second site (Nura River site), it can be concluded that the acetaldehyde production plant, which was closed more than 20 years ago, left a significant negative mark on the ecosystem of Central Kazakhstan and the public health of the local population in the form of Hg pollution of Nura River and Samarkand and Intumak reservoirs. The contamination was significant and mainly localized, with the majority of Hg deposited in topsoils and riverbank deposits within 25 km from the discharge point.

The first 10–20 km of the riverbed from the wastewater discharge point was confirmed to be the most contaminated part of the river in terms of Hg in soil, sediments, and technogenic silts.Special attention should be paid to the river sediments that may contaminate water during annual floods.The local population might be exposed to Hg due to the consumption of locally caught fish from the river and reservoirs; hence, it is necessary to prohibit fishing and consumption in the region.

It is necessary to mention a deficit of the relevant recent follow-up studies, so further research of the area is recommended to verify the previous findings and assess the current situation. Overall, the review of available literature showed significant gaps in terms of recent investigations and a lack of integrated site assessment. Moreover, both sites seem to be in need of further remediation action. The suggested potential remediation responses include removing contaminated silts from the Nura riverbanks whereas multiple alternatives seem to be present for Lake Balkyldak from soil washing to electrokinetic remediation. However, the recommendations are preliminary, and these techniques need to be validated based on the latest field data and on pilot-scale laboratory experiments. It has been established that cities and surrounding territories of Temirtau and Pavlodar are still Hg-contaminated regions because of the persisting impact of former acetaldehyde and chlor-alkali plants. As the literature is fragmented, fails to provide complete information on Hg pollution in the regions, indicates the persistence of contamination, and presents data which is now outdated, a comprehensive site characterization effort is necessary to establish an up-to-date picture of the current situation for both sites.

## Figures and Tables

**Figure 1 ijerph-17-08936-f001:**
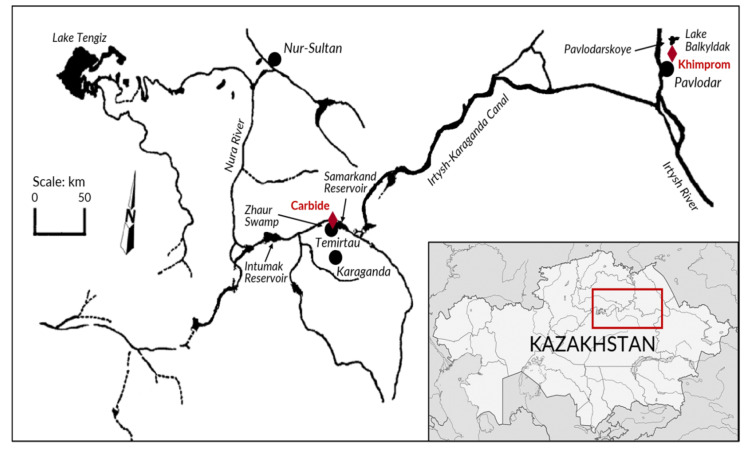
Map showing Hg-contaminated sites in Kazakhstan (adapted from [22]).

**Figure 2 ijerph-17-08936-f002:**
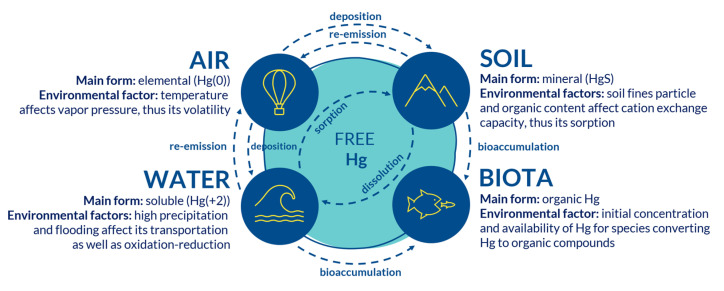
Main forms of Hg and environmental factors affecting Hg presence in different media.

**Figure 3 ijerph-17-08936-f003:**
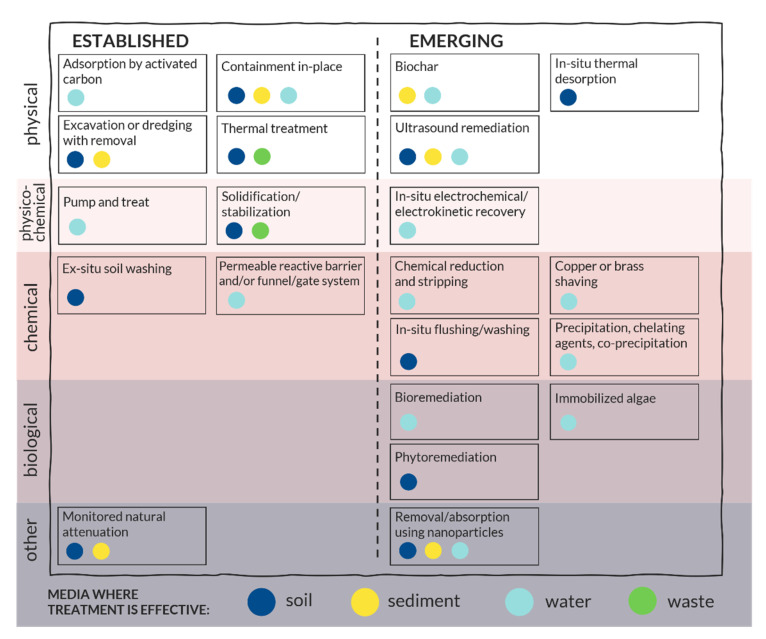
Established and emerging remediation technologies for Hg-contaminated zones.

**Table 1 ijerph-17-08936-t001:** Maximal permissible concentrations of mercury.

Medium	World Health Organization	USA	Canada	E.U.	Kazakhstan
In air within working zones (μg/m^3^)	not available (n.a.)	100 (8 h and ceiling) [29]	(n.a.)	elemental and inorganic, Sweden: 30 (8 h) organic: 10 (8 h) [5]	5 [35]
In ambient air of populated areas (μg/m^3^)	1 (annual) [36]	(n.a.)	(n.a.)	(n.a.)	0.3 [35]
In water for sanitary and domestic use (μg/L)	1 [14]	2 (maximum contaminant level) [37]	1 [38]	drinking water: 1 (parametric value) [39] surface waters: 0.07 (maximum contaminant level) [39]	0.5 [40]
In soil for agricultural use and in residential places (mg/kg of soil)	(n.a.)	California: 1 [41]	6.6 [42]	Netherlands: 0.83 [43]	2.1 [44]
In soil of other areas (mg/kg of soil)	(n.a.)	industrial, California: 4.4 [41]	industrial: 50 [42]	industrial, Netherlands: 4.8 [43]	10 [44]
In biota (mg/kg wet weight)	fish: 0.5 (MeHg) predatory fish: 1 (MeHg) [45]	1 (MeHg) [45]	all fish except shark, swordfish and tuna: 0.5 [45]	fish: 0.5 predatory fish: 1 [45]	fish: 0.3 predatory fish: 0.6 [46]

**Table 2 ijerph-17-08936-t002:** Summary of literature on Hg contamination in Kazakhstan.

Study	Title	Objective	Sampling	Analysis	Main Findings	Conclusions	Recommendations
2000a Heaven et al. [54]	Mercury in the River Nura and its floodplain, Central Kazakhstan: I. River sediments and water	Establish the location, extent, and nature of the contaminated sediments and evaluate the potential for sediment transport in the Nura river	River sediments (n = 156); water (n = n/a); Surveying of the riverbed and backwaters with sediments thickness	Sediments: acid digestion + CV-AAS; Water (on site): Au pre-concentration and SnCl_2_ reduction + portable AAS	Sediments: very high concentrations in first 15 km downstream, average 150–240 mg/kg; Estimated total silts between Temirtau and Intumak Reservoir (75 km) 463,500 m^3^ or 9.4 t Hg; A major part of polluted sediment not transported far downstream (except for floods) Hg in backwaters reflect changes of Hg in riverbed silts, in first 5–15 km average 86 mg/kg, falling to 2.6 mg/kg near Intumak Water: mean <1 μg/L (EU limit for inland surface waters & WHO (1984) in drinking water); but > 0.3 μg/L local limit for fish-inhabited waters; No general relationship between amount of suspended particulate matter (SPM) and total, dissolved or suspended Hg	Results were lower than expected; most of the contaminated silts do not appear to be readily transported for long distances downstream (also confirmed by hydraulic modeling work (not presented)); Hg in the water leaving the reservoir would suggest that more than 100–200 kg Hg/year could be moving downstream (mainly dissolved form)	The desired option for reclamation is to dredge silts from the outfall canal and remove highly polluted sediments from its banks; dredge technogenic silts from the first 25–30 km of riverbed below Temirtau and limit further distribution; remove silts with >10 mg/kg Hg deposited on banks Management of discharge: to reduce flood size and prevent disturbance of sediment
2000b Heaven et al. [55]	Mercury in the River Nura and its floodplain, Central Kazakhstan: II. Floodplain soils and riverbank silt deposits	A detailed survey of the floodplain to investigate the extent of pollution and to assess the need for remediation	A survey covering 160 km2 of the floodplain of River Nura (72 lakes in total); Topsoil samples 0–15 cm (n = 1100); Silts at highly-contaminated Zhaur Swamp (n = 157 from 28 boreholes); Additional soil samples from irrigated areas in 1998 (n = 10)	Acid digestion + AAS Silts—preliminary sequential extraction tests: 0.1 M HCl	Topsoils (53 t Hg): from 0.01 to >100 mg/kg, Hg > 21 mg/kg (tenfold local limit) mainly in first 25 km of river, > 10 mg/kg (Dutch intervention value) up to 60 km downstream, River bank deposits/silts (65 t Hg): mean 73.3 mg/kg in most contaminated section; mean 13.4 mg/kg 70 km downstream Zhaur swamp formerly used as waste disposal area (62 t Hg): up to 1974 mg/kg at the surface; fall rapidly with increasing depth, mean total Hg 306.7 mg/kg in upper 20 cm In older silts w/total Hg 10.2–10.7 mg/kg: highly mobile 71.6–87.9%, rel. mobile (oxides) 2.1–3.5%, insoluble 7.2–24.6%; In sediment from riverbed below Intumak with total 0.017 mg/kg: highly mobile 15.5%, relatively mobile 10%, insoluble 74.5%	The contamination is severe but relatively localized, with >70% of mercury in topsoils and >90% of mercury in riverbank deposits located within 25 km from the source.	Removal of the silt deposits from banks in the first 30 km below outfall would remove >90% of Hg; isolate Hg by stockpiling silts under a meter of inert cover material in a location safe from groundwater intrusion and flooding; cease cultivation of Zhaur Swamp; remove and isolate upper 40 cm of soil; soils with > 10 mg/kg should be taken out of agricultural production; minimize flooding of contaminated areas of the Nura valley by regulating the discharge from Samarkand Reservoir
2000 Yanin [56]	Mercury in the epiphyte retained of the Nura River (Kazakhstan) as an indicator of technogenic pollution.	To evaluate the effectiveness of epiphytic suspension in assessing the level and scale of waterbodies pollution by mercury	Epiphytic suspension from *Myrio phyllum specatum* L. (dried and separated) Technogenic silts (dried and sieved)	AAS (IMGRE-900 mercury analyzer)	Maximum total Hg concentrations in technogenic silts near wastewater discharge (about 6–10 km); 0.05 km from outfall mean total Hg in silts 33.54 mg/kg, 10 km away mean Hg is maximum 47.62 mg/kg; Hg in technogenic silts > Hg in epipihtytes; epyphytes can be used as an indicator	Epiphyte suspension (which intensively concentrates Hg) reflects the influence of various Hg sources to watercourses and shows the extent of pollution. Hg in technogenic silts is mostly deposited in the vicinity of the wastewater discharge	It is proposed to use epiphytic suspension, i.e., suspension precipitated on macrophytes, to estimate the level and scale of the technogenic pollution of Hg’s rivers.
2007b Ullrich et al. [57]	Mercury distribution and transport in a contaminated river system in Kazakhstan and associated impacts on aquatic biota	To investigate the transport, fate, and bioavailability of Hg in the Nura river system by analyzing sediments, water, plants, and fish sampled from the river system	Sediments from different years, locations, and depths along the river; water, unfiltered and filtered (0.45 μm); plants (cattail and reed); fish from the river, lakes, local market (n = 130, 20, 6)	Sediments: acid digestion + CV-AAS (Perkin-Elmer AAnalyst 100), acid digestion + CV-AFS, MeHg—modified Westöö procedure, GC-ECD; water: total Hg and suspended solids—BrCl, SnCl_2_ reduction + CV-AFS (Millennium Merlin); plants: acid digestion + CV-AAS (), CV-AFS; fish: acid digestion + CV-AFS	Sediments within 20 km downstream of effluent—highly polluted, a strong source of water contamination; THg in most contaminated section = 9.95 to 306 mg/kg; Highest MeHg in surface sediments (4.9–39 ug/kg) <0.1% THg; the significant inverse relationship between THg and MeHg% formed in sediments Unfiltered surface water during flood peak THg = 1600–4300 ng/L Background concentrations of Hg in surface water are not reached for 200 km downstream, even in wetlands during flood In aquatic plants Hg in most contaminated section = 15–20x background; fish impacted for >125 km downstream from the source—significant transport of dissolved MeHg to downstream areas, in situ MeHg production	Elevated Hg concentrations in water, fish, and aquatic plants near impoundments appear to indicate that Hg’s availability for methylation may be increased in these areas. The high immobilization of mercury by industrial sludge, the basis of which was the ash of a thermal power station, makes debatable the rationale for cleaning up the mercury-containing bottom sediments of the Nura River under the project of the International Bank for Reconstruction and Development.	Studies on terminal wetlands of the Nura, methylation capacity at Intumak and Samarkand barrage Prevent further transport of Hg to downstream reaches
2010a Hsiao et al. [3]	Burdens of mercury in residents of Temirtau, Kazakhstan I: Hair mercury concentrations and factors of elevated hair mercury levels	To evaluate Hg exposure levels through concentrations in hair of the local population; to describe the relationship between Hg concentrations in hair and dietary intake and other factors; to identify group at high risk of Hg exposure	Hair from Temirtau and Almaty (n = 289 and 13), fish purchased or caught locally (n = 111), food (veg, milk, beef) (n = 24)	Hair: Rigaku Mercury Analyzer SP-3 or MA-2; fish, food: acid digestion + CV-AFS (Millennium Merlin)	Hg in hair = 0.009−5.184 μg/g, mean 0.577 μg/g; in ~17% of population >1 μg/g A positive correlation between Hg in hair and frequencies of river fish consumption Subgroups of males, people >45 y.o and fishermen or anglers—elevated levels	The mean concentration of Hg in the river fish being 0.43 μg/g and an average bodyweight of 67 kg of the local people; Hg in hair at a moderate level, exposure levels not very severe	Raise awareness of the dangers of consuming fish caught in River Nura and its oxbow lakes below Temirtau, or at least decrease consumption rate to no more than once a week, especially for pregnant women
2010b Hsiao et al. [21]	Burdens of mercury in residents of Temirtau, Kazakhstan. II: Verification of methodologies for estimating human exposure to high levels of Hg pollution in the environment	To evaluate the exposure risk posed by Hg waste from a disused acetaldehyde plant at Temirtau, to identify the adaptability of these approaches, and discuss the uncertainty and variability generating in the methodologies of exposure assessments	Fish (n = 21), food (vegetables, milk, beef) (n = 24), soils (n = 27), loose dust (n = 38), hair (n = 289); questionnaire (n = 232)	Fish, food: acid digestion + CV-AFS (Millennium Merlin), soil and dust: acid digestion + CV-AAS (Perkin-Elmer AAnalyst 100), hair: Rigaku Mercury Analyzer SP-3 or MA-2; HQ = Average daily intake/RfD	Probabilistic (Monte-Carlo): ADD of MeHg mean 0.08 (0.003–12.233) μg/kg body weight/day—75% of MeHg intake via fish; 19% population exceeded 0.1 ug/kg BW/day Non-carcinogenic risk due to MeHg contamination: HI = 0.36 at 50 percentile, but HI = 2.53 at 95 percentile Deterministic: HI = sum of HQ = 7.92, where HQ = 7.62 from fish consumption	The probabilistic approach (MC simulation) is slightly overestimated, but the stable and reliable prediction for the high-end exposed population, while the deterministic approach overestimated ADD 1.5 times than values derived from hair. Fish and shellfish consumption—major route of MeHg exposure	Probabilistic approach robust, useful, and reliable in assessing accurate levels of exposure to Hg
2002 Ilyushchenko et al. [58]	Mercury (Hg) contamination of fish fauna of Balkyldak technical pond	To investigate the extent of mercury contamination in the fish fauna of the Balkyldak lake	Fish from Balkyldak (n = 55): tench, common perch, silver crucian carp, Siberian dace	Acid, bromide-bromate digestion + CV-AFS (PSA 10.025 Millennium-Merlin)	In 50 out of 55 Hg in muscle tissue > 0.3 mg/kg (max allowable concentration in dace/crucian carp/tench) Average total Hg = 4.36/3.18/1.98	Limited sample size does not allow to draw exact conclusions	Further research of Hg accumulation in other aquatic organisms and Hg migration along the food chains of the ecosystem of Balkyldak (hydrobiological and trophological research methods)
2004 Woodruff and Dack [19]	Analysis of risk from mercury contamination at the Khimprom Plant in Kazakhstan	To examine mercury contamination at the chlor-alkali plant at Pavlodar and to establish whether risks to human health exist from this contamination via vegetable consumption and soil ingestion	Surface soils from site and Pavlodarskoye village, groundwater (n = unknown)	Risk assessment (ingestion of food and soil): (1) UK Contaminated Land Exposure Assessment (CLEA) model; (2) The Netherlands, Van Hall Institute Risc-Human Model Version 3.0	Hg in soil from plant far and close to contaminated zones (1997-8 and 2001-2) = 0.0067 and 835.9 mg/kg respectively; Hg in soil from Pavlodarskoye village (2001-2) = 1.5 mg/kg; Hg in groundwater from plant (1997-8 and 2001-2) = 0.00022 and 18 mg/L, respectively; Hg in groundwater from Pavlodarskoye village (2001-2) = 0.005 mg/L; CLEA model—vegetable uptake and ingestion of soil on vegetables; risk not related to increased Hg concentrations in the latter years Risc-Human model—ingestion of soil by children, ingestion of meat and vegetables children and adults	Possible health risks to the local population at Pavlodarskoye from the consumption of homegrown vegetable uptake and with ingestion of soil attached to homegrown vegetables	To obtain a more representative value of calculated risk, factoring in these differences recommended that vegetable uptake be studied further
2005 Ilyushchenko et al. [59]	Activities for prevention of the threat of river Irtysh mercury pollution in Pavlodar, Kazakhstan	Report on the results and strategies for preventing pollution	-		Four large hotspots with Hg in soil = >500 × 2.1 mg/kg 2931 kg of Hg for the industrial site No.1, 16,022 kg of Hg for the area between the industrial site of former PO “Khimprom” and lake Balkyldak; In the tissue of fish from the lake 0.18-2.2 mg/kg The plume of Hg-contaminated groundwater: up to 150 ug/L, decreasing with distance from hotspots In surface water: (i) atmospheric precipitations in lagoons ≤50 mg/L; (ii) surface water to the south from lagoons 3–30 ug/L; (iii) surface water in a ditch along lake 2–18 ug/L; (iv) surface water of lake 3.4 ug/L (near lagoons) to 0.1–0.3 ug/L (along the rest of the shore) In unfinished emergency canal from the lake to west ≤0.01 ug/L; in Irtysh river <0.002 μg/L; in oxbow lakes near village <0.009 ug/L; in predatory fish from river 0.075–0.16 mg/kg	Four scenarios of Hg transport with groundwater until 2030: (1) direction of plume does not change—no severe threat to village and river; a limited amount of Hg might enter the emergency canal; (2) cut-off wall around building 31; (3) containment of both sources of pollution—eliminate groundwater contamination; (4) changes of hydrogeological conditions in the northern industrial area of Pavlodar depending on industrial development or degradation	Instead of expensive and ineffective recovery of Hg from highly contaminated wastes, the containment strategy was proposed assuming isolation of major hotspots from the atmosphere, surface run-off, and groundwater. A cut-off wall built-in 2003–2005 around four major hotspots; topsoil to 0.5 m excavated and removed to isolated sites; building 31 demolished; monolith storage
2006 Panin and Geldymamedova [60]	Ecological and geochemical characteristics of soils in Pavlodar, Republic of Kazakhstan	To identify the presence of heavy metals and other chemical elements in soils of Pavlodar city	Soil from Pavlodar and surrounding areas (n = 609)	Acid digestion + AAS (Perkin Elmer 403 + HGA-74)	Hg in the city was in the range of 0.08–18.96 mg/kg Mean total Hg in northern industrial zone 3.51 mg/kg, in the north part of the city 0.21 mg/kg Mean concentration of elements 1.6–22.5 times higher than background (especially Hg, Cd, Co, Mo) Northern industrial zone: max Hg, V, Sr, Ni compared to other areas Zc > 128 (ecological disaster area) on the territory of chemical plant	Maps of the distribution of chemical elements in the soils are compiled. The highest concentrations of chemical elements in the northern industrial zone	
2007a Ullrich et al. [50]	Mercury contamination in the vicinity of a derelict chlor-alkali plant. Part I: Sediment and water contamination of Lake Balkyldak and the River Irtysh	To investigate the impact of Hg emissions from the chlor-alkali plant on the surrounding environment and, in particular, the lake (sediments, water, and biota)	Sediments (n = 55) and water from Balkyldak (n = 38); sediments and water from Irtysh (n = 32), water from oxbow lakes (n = 18); soil from 6 locations around lake	Acid digestion + CV-AAS, CV-AFS	Hg in sediments in the lake, near wastewater outfall pipe up to 1500 mg/kg; Hg in lake water in the range of 0.11–1.39 ug/L (mainly in the southern part); on windy days, concentration up to 7.3 ug/L; Hg in river sediments up to 0.046 mg/kg in the old river channel & up to 0.36 mg/kg in floodplain oxbow lakes; Hg in river water—not detected, in oxbow lakes—trace (3-9 ng/L) Hg in soil around lake—2.65 mg kg^−1^ (0.22–5.72 mg kg−1) at 0–10 cm depth, 1.81 mg kg^−1^ at 10–20 cm, and 1.14 mg kg^−1^ at 20–50 cm	Balkyldak sediments are heavily contaminated. Thus, the lake poses a threat and needs remediation; Hg does not significantly impact the Irtysh river A cut-off wall around lagoons and clay cover eliminated a major source of Hg to the lake. However, the lake still receives Hg via an old outfall pipe	Recommendations: ex-situ dredging or disposal; thermal desorption; capping and dredging Preliminary tests carried out on a sediment sample taken from the south of Lake Balkyldak indicated that most Hg was present as elemental Hg, and >95% of the Hg could be removed by ultrasound (unpublished data).
2007c Ullrich et al. [61]	Mercury contamination in the vicinity of a derelict chlor-alkali plant Part II: Contamination of the aquatic and terrestrial food chain and potential risks to the local population	To gain a preliminary insight into the potential for contamination of the terrestrial food chain and the associatedlevel of risk.	Water from lake Balkyldak (n = 55), from Irtysh and oxbow lakes (n = 30), water from wells (n = 30); Cow milk from village (n = 15), liver and kidney from 1 cow; soil (n = 24)	Water: acid digestion + CV-AFS	Fish from Balkyldak seriously contaminated by Hg (dace>carp>tench) Mean (range) Hg in perch from Balkyldak 0.89 (0.16–2.20 mg/kg) >>> in perch from Irtysh 0.112 (0.075–0.125 mg/kg) (limited sample size) Hg in 91% of fish exceed the permissible level Mean (range) Hg in soil = 1.04 (0.10 and 3.30) mg/kg Hg in groundwater < LoD (5 ng/L) Hg in bovine milk samples < 2 ug/kg, tissue = 10.96 ug/kg	Hg in fish from lake Balkyldak exceed current human health limits; so, consumption of contaminated fish appears to be the main route of exposure for humans	Eliminate the current fish population by using rotenone (fish poison) Environmental and human health impacts associated with cattle grazing on contaminated land around the plant and drinking contaminated surface waters To investigate Hg uptake from vegetables grown in contaminated soil
2009 Shaimardanova et al. [62]	Heavy Metals Accumulation in Children Hair	To justify the accumulation rate of chemicals (Hg, Zn, Se, Rb) in children’s hair living in Pavlodar as a method of an environmental assessment of the quality of urban ecosystem under conditions of long technogenic impact	Children hair 12–14 y.o. (n = 100)	Instrumental neutron activation analysis (INAA)	Highest Hg accumulation in W, SW, NW districts = 0.5–0.7 mg/kg due to proximity to the industrial zone Uneven distribution of toxic elements in human biosubstrates	High Hg and Zn content due to high mobility in “soil–snow– plants–biosubstrates (hair)” system Two groups of main exposure sources: coal energy and metal industry (Hg, Zn); chemical (Hg, Se, Rb) and petrochemicals (Zn, Se); Most contaminated districts: NW, W, SW	
2011 Ilyuchshenko et al. [63]	Final technical report	(1) Risk assessment on the flow direction of groundwater polluted with oil products and Hg, including its passage through sampling wells in Pavlodarskoye village, joining River Irtysh and/or resurfacing at pastures. In case of high risk, building strategy to control and minimize it; (2) Building risk management strategy for the environment from Hg pollution of Lake Balkyldak, including pollutants bioaccumulation in the food chain.	Surface and ground water (n = 800), bottom sediment (n = 334), soil (n = 610), grass (4 g), biota from Balkyldak (n = 132); water for MeHg (n = 3), Hydrogeologic modeling for risk assessment and management of groundwater pollution via ModFlow GMS 5.0. Sampling plan of top 3 soil layers (0-10, 10-20, 20-50 cm) in a regular grid with a varying sampling step	AAS (Lumex RA 915+); AFS (PS Analytical Millennium Merlin System)	Computer model of Hg contamination of groundwater verified by Hg analysis -> predicted Hg transport within 30 y Hg plume in groundwater continues to spread in N-NW direction of the plant -> high risk of pollution of topsoil layers Hg levels in GW in the eastern direction are falling as predicted by computer modeling Hg levels on plant’s territory are unpredictable, but in topsoils of most parts are very high despite demercuration efforts Waste storage facilities of PCP show good isolation results The estimated amount of Hg discarded in Lake Balkyldak by the plant—135,336 kg Hg levels of surface waters of Balkyldak decreased after the burial of Hg waste Hg levels of biota fell as well, but some fish specimen had high Hg levels in 2007 (1–1.5 mg/kg)	Topsoils and vegetation: >2.1 mg/kg (MPC) in selected sites with Hg-bearing groundwater; in soils on PCP site extremely high—up to x1000 MPC Groundwater: extremely irregular decrease; high risk of formation of new hotspots of soil contamination on PCP territory due to the transport of soluble Hg to aeration zone; no risk of the Irtysh and water-supply wells of Pavlodarskoye village contamination if hydro-geological conditions remain same;	(1) Create a monitoring laboratory for PCP to complete implementation of post-demercuration monitoring programs (2) Treatment for soils: pulpation + gravitational separation (3) Bioremediation of groundwater to immobilize Hg (4) Sediments: pump using a dredger and move to an isolated pond with subsequent evaporation and burial (5) Ban the consumption of fish from Balkyldak (6) 2nd phase of demercuration plan to address: (i) treatment and remediation of soils on plant’s territory (ii) Hg immobilization in groundwater (iii) treatment of Balkyldak’s sediments
2016 Shakhova et al. [64]	Evaluation of mercury contamination in the vicinity of enterprises of the petrochemical complex in the winter period (based on the example of Pavlodar, Republic of Kazakhstan)	To evaluate mercury pollution in the vicinity of petrochemical complex enterprises during the winter period (on the example of Pavlodar) according to the study of/investigating/analyzing the snow cover as storage of solid particles.	Snow from 11 locations (1 sample of 10-12 L), number of samples at the closest residential area = 5	AAS (Lumex RA 915+ and PYRO 915)	(1) Hg concentration in solid fraction of snow exceeds max allowable concentration by 1.5–7 times. In NE zone: 0.31–1.04 mg/kg, SW: 0.22 mg/kg, NW: 0.03–0.26 mg/kg, background 0.15 mg/kg; in Pavlodarskoe village close to background Daily mean Hg deposited on snow cover 4.9–221 mg/(km^2^ × day); max—1.5 km from the plant, NE zone	High Hg concentrations in the NE zone might be related to technogenic Hg contamination and wind directions; Hg depositions and concentrations in snow covers are high (0.03–1.04 mg/kg) in the vicinity (0.5–2.5 km) from PCP	The data obtained can be used for planning of environmental activities, such as air monitoring in the northern industrial zone of Pavlodar, as well as for further monitoring of health risks of the Pavlodar region population

**Table 3 ijerph-17-08936-t003:** Comparison of two cases of Hg contamination in Kazakhstan.

Case	Pavlodar (Includes Balkyldak Lake and Irtysh River)	Nura River
Source of contamination	Chlorine and caustic soda production at the JSC “Pavlodar Chemical Plant,” Hg used in electrolysis	Acetaldehyde production at the Temirtau chemical plant by direct C_2_H_2_ hydration in the presence of HgSO_4_
Years of operation	1975–1993	1950–1997
Contaminated zones	The territory of the plant, lake Balkyldak for wastewater discharge, Shoptykol	Nura river (mainly close to the discharge point), Intumak and Samarkand reservoirs, Swamp Zhaur, old ash lagoon of KarGRES-1 and wastewater treatment facilities, terminal wetlands of Korgalzhyn National Park
Hg quantities discharged	1300 t [65]; 1000 t [50]; 19 t on the territory of the site [59]; discarded into lake 135 t [63];	2351.6 tons of Hg consumed [19]
Hg concentrations in soil and groundwater (mg/kg for soils, mg/L for water)	Soils: Max: 835.9 mg/kg [19], >1000 mg/kg [59], 2000 mg/kg [63]; Topsoil near Balkyldak: mean 2.65 mg/kg [50]; In the city and PCP (2001–2002): mean 3.51 mg/kg, in city 1.8 mg/kg [60]; in the village: 1.04 and 1.5 mg/kg [57] and [19]. Groundwater: up to 150 μg/L [59], <5 ng/L [57]	Soil: Local hotspot (Swamp Zhaur): mean 306.7 mg/kg [66]; In Nura floodplain and banks between Samarkand and Intumak reservoirs: 0.01–100 mg/kg, mean 5.9 mg/kg 12.5 km downstream, 10 mg/kg 60 km downstream [55]
Hg concentrations in water (mean)	0.11–1.39 μg/L with an increase up to 7.3 μg/L on windy days [50] undetectable and trace levels [50,59].	Surface water in flood periods: 1.6–4.3 μg/L [61], 0.5 μg/L to 1.25 μg/L [54] During other seasons−usually<0.5–1 μg/L, sometimes >0.3 μg/L [54,66] Intumak reservoir and further terminal wetlands in the Korgalzhyn National Park <5 ng/L [54,61]
Hg concentrations in sediments (mean)	wastewater outfall pipe—167 mg/kg, old river channel > 0.050 mg/kg [50] more distant locations—40–60 mg/kg [50]	The highest concentration (150–240 mg/kg)—within the first 15 km [54] 9.95–306 mg/kg in the most polluted section [61]
Hg in air, biota, hair, blood (mean)	Fish from Balkyldak (2001): 0.18–2.2 mg/kg [58,59]; perch 0.89 mg/kg [57]; Fish from Irtysh and oxbow lakes (2001): 0.075–0.16 mg/kg [59], 0.112 mg/kg [57]; Fish (2007): decreased locally, 1–1.5 mg/kg [63]; Plants 1–1.5 km away (2001): 1.09–1.66 mg/kg [57]; Vapor: 100–1600 ng/m^3^, max >10,000 ng/m^3^ [63]; In snow sediments 0.5–2.5 km from PCP: 0.03–1.04 mg/kg [64] Pavlodar children hair: 0.44 ± 0.5 mg/kg, in children from western districts 0.5–0.7 mg/kg [62] Bovine milk Pavlodar: < 2 μg/kg, in cow tissue—10.96 μg/kg [57]	River fish (2002): 0.325–0.923 μg/g [3]; crayfish caught and bought from local market: 0.043 and 0.026 mg/kg [61]; The narrow-leaf cattail from riverbanks (0.03–0.63 mg/kg) Temirtau region’s citizens’ hair—0.009–5.184 μg/g [3]

**Table 4 ijerph-17-08936-t004:** Comparison of selected international cases.

	Facility	Years of Operation	Production Capacity	Estimated Discharge	Hg in Soils (mg/kg)	Hg in Sediments (mg/kg)	Hg in Water (ng/L)	Main Remarks	Reviewed References
Temirtau, Kazakhstan	Acetaldehyde	1950–1997	Not reported in reviewed references	1200 t [67]	0.01 to over 100 [55] up to 1974 [66]	150–240 [54] 9.95–306 [50]	500–1250 [54] 1600–4300 [50]	Reviewed in detail in the present study (Refer to Table 3).	[50,66,67]
Schkopau, Germany	Acetaldehyde	Before 1942–1989	142,800–300,000 t/y	Not reported in reviewed references	14 to over 1000 [68]	Not reported in reviewed references	Not reported in reviewed references	An industrial complex with three chlor-alkali plants and one acetaldehyde plant [68]; A strong source of atmospheric Hg emissions (0.2–1.7 kg/day) [69]; High levels of organic Hg compounds in the plant’s soils, possibly due to reaction of Hg^0^ with carbide (CaC_2_) dust covering the site’s soils, reaction of Hg with soil’s humus, intermediate reactions in acetaldehyde production [68].	[68,69,70]
Minamata, Japan	Acetaldehyde	1932–1968	Not reported in reviewed references	250 t [71]	Not reported in reviewed references	Before dredging: up to 2000 [71] After dredging: 0.49–3.4 [72] 0.61–6.73 [71]	1.3–4.3 (total [71]) 0.10–0.95 (dissolved [73])	>2200 registered cases of Minamata disease in the area by 2003 [72]; Dredging in 1974–1990 to decrease sediment Hg below 25 mg/kg [72]; Hg in sediments still 70 times higher than background levels [71]; 254% of total Hg in the bottom water layer is MeHg -> MeHg is the main form of Hg from sediment to water [71].	[71,72,73]
Qingzhen, Guizhou, China	Acetic acid and acetaldehyde	1971–2000	Not reported in reviewed references	134.6 t [74]	14.3+−0.1 to 354+−15 [75] 0.06–321.38 [74] 0.14–259.56 [76]	Not reported in reviewed references	450–1830 (total) 12.7–16.1 (dissolved [75]	Site conditions similar to Nura River; Total daily intake in μg/kg of 60 kg body weight for rice (inorganic Hg—0.001–0.003, MeHg—0.005–0.019) irrigated by polluted river water and fish (inorganic Hg—0.000–0.003, MeHg—0.01–0.13) were estimated and exceeded values recommended by U.S. EPA [75].	[74,75,76]
Pavlodar, Kazakhstan	Hg-cell chlor-alkali plant	1975–1993	100,000 t of Cl_2_/y	1000–1300 t [50,65]	0.0067–835.9 [19] 0.22–5.72 (around the lake, excluding the plant’s site, [50]) up to 2000 [63]	0.11–617 [50]	110–7300 [50]	Reviewed in detail in the present study (Refer to Table 3).	[19,50,63,65]
Penobscot River, Maine, US	Hg-cell chlor-alkali plant	1967–2012	65,700 t of Cl_2_/y	9 t	Not reported in reviewed references	0.35–1.10 [77]	2 (dissolved [77])	Slow drainage of Hg from the site into the river (5.4 g/day) with increased loading during major storm events [78]; site groundwater input of Hg amounted to 17 g/day in the 1990s and decreased to 0.11 g/day in 2009 due to capture and treatment of site groundwater [78]; slow recovery of river’s aquatic food web, need for longer monitoring [79].	[77,78,79]
Flix, Spain	Hg-cell chlor-alkali plant	1949–2017	115,200 t of Cl_2_/y	Not reported in reviewed references	0.044–12.9 [80] 1.7–61.6 [81]	0.098–495 [80] up to 640 [82]	Not reported in reviewed references	The current pollution source is a sludge deposit formed at the riverbank close to the dam and containing approximately 10-18 Mg of Hg [83]; Several dams placed upstream and downstream of the plant resulted in accumulation of waste containing Hg at the base of the downstream dam in the absence of natural dilution and burial with river sediment material [82]; removal of the contaminated deposit has started [81,82]. Elevated atmospheric Hg levels in the region (229 ng/m^3^ in the vicinity of the plant) exceeding guideline thresholds for a residential area in Flix town [80].	[80,81,82,83]
Estarreja, Portugal	Hg-cell chlor-alkali plant	1950–2002	Not reported in reviewed references	over 50 t	0.18–49.23 [84] 0.01–90.8 [85]	Up to 180 [84] 0–48 [86]	7-84 (dissolved [86]) 12–847 (total [85])	About 8 km^2^ area around the plant has been identified as a heavily contaminated zone [84]; Hg tightly binds to topsoils with a size fraction <0.063 mm, possibility of groundwater contamination [84]; Hg concentrations in surface sediments were lower than those in deeper parts, indicating higher historical Hg releases with pieces of evidence of remobilization of Hg in water in reducing environments [86]. Agricultural soils located close to the plant’s past effluent discharge spot contain high Hg levels and recommended to be restricted from use and remediated [85].	[84,85,86]
Dalhousie, New Brunswick, Canada	Hg-cell chlor-alkali plant	1963–2008	34,300 t of Cl_2_/y	141–163 t (2 chlor-alkali plants of Canada [87])	Not reported in reviewed references	<0.1–8.1 [88] 0.02–1.96 [88] 0.04–0.28 [89]	840–4320 (total, in effluents [88])	Atmospheric Hg emissions from the facility in 1988–1996 were in the range 31–70 kg/y, with 2.5 times higher Hg quantities discharged in the facility’s landfilled sludge [90]; The sludge pile close to the plant contains approximately 2.5 μg of Hg/g in the form of Hg sulfides [90]; Hg content in sediment samples close to the plant exceed Canadian guideline values (0.13 mg/kg) [88]; Recent observations indicate natural recovery due to sediment deposition [89].	[87,88,89,90]
Rm Valcea, Romania	Hg-cell chlor-alkali plant	1968-present [91]	210,000 t of Cl_2_/y	36–53 t (estimated based on data from [91])	Not reported in reviewed references	0.5–45 [91]	9–88 (dissolved [92])	Fluctuations in Hg concentration in sediment cores -> flooding cause the transport of contaminated soil from the site to the river sediments [91]; Bravo et al. [91] strongly discouraged any treatment actions involving rework or dredging of the sediments due to the risk of resuspension of buried highly contaminated particles.	[91,92]

**Table 5 ijerph-17-08936-t005:** Selected reviewed remediation technologies.

Remediation/Treatment Technology	Media	Description	Removal Values	References
Adsorption by activated carbon	Water	A universal adsorbent material to reduce flux to the environment	60–95%	[6,100,108]
Biochar	Sediment, Water	Sorption of the contaminant by biochar—charcoal produced from plant matter	up to 95% in pore water	[48,102]
Bioremediation: bio-treatment, biofunctionalized zeolite, genetically engineered bacteria	Water	A process that generally utilizes microorganisms, plants, or their enzymes to decrease the toxicity of the contaminant	91–95%	[6,109,110]
Chemical reduction and stripping	Water	Injecting chemically reductive additives into contaminated media/chemical reductant in the contaminant plume; physical separation from the aquatic stream by vapor	>94%	[100,102,111]
Containment in-place	Soil, Sediment, Water	Covering contaminated media with clean soil and/or other low permeability material	not applicable	[6,112]
Copper or brass shavings	Water	Removal of Hg_2+_ from water by the amalgamation	96–98%	[48,113,114]
Ex situ soil washing	Soil	Washing the excavated soils with a special solution, scrubbing, and separating clean soil	up to 99%	[6,100,115]
Excavation or dredging with removal	Soil, Sediment	Removal and off-site storage of the contaminated material	not applicable	[48,116]
Immobilized algae	Water	Accumulation of the contaminant from aquatic media in certain species of algae	up to 90%	[100,102,117]
In situ thermal desorption	Soil	On-site heating soil to very high temperatures to release contaminant in gaseous/vapor phase	99%	[6,112,118]
In situ flushing/washing	Soil	Flooding a zone with a flushing solution to mobilize contaminant	35–90%	[50,112]
In situ electrochemical/electrokinetic recovery	Soil	Applying low-intensity direct current across electrodes to drive ions migration to the opposite sign electrode using a mobilizing solution	30–92%	[102,112]
Monitored natural attenuation	Soil, Sediment	Natural physio-chemical/biological processes reduce concentration/toxicity/mobility	not applicable	[6,116]
Nanotechnology	Soil, Sediment, Water	Injected FeS nanoparticles to contaminated soil immobilize Hg via ion exchange/adsorption	~92% Ag-Zn > 99%	[102,109,112]
Permeable reactive barrier and/or funnel/gate system	Water	A subsurface construction used to channel the contaminated plume into a gate with reactive material to adsorb/decompose/transform the contaminant	variable (material- and site-specific)	[119]
Phytoremediation (phytostabilization, phytoextraction, phytovolatilization)	Soil	A process that uses plants to remove, stabilize, or destroy the contaminant	>99%, 2.62 mg/kg max removal efficiency	[100,103,116,120]
Precipitation, co-precipitation, chelating agents	Water	A chelating reagent is added to the contaminated water in soluble form, and the contaminant is removed after its flocculation and/or precipitation	variable (reagent-specific)	[111,121]
Pump and treat	Water	Pumping contaminated groundwater to treatment system, discharging back to environment	variable (technology specific)	[112]
Solidification/Stabilization	Soil, Waste	Physically encapsulating or chemically stabilizing the contaminant in the soil	90–98%	[6,109,112]
Thermal treatment: (1) batch retorting, (2) ex situ thermal desorption, (3) vitrification	Soil, Waste	(1) Heating contaminated material under vacuum to volatilize Hg volatilization, (2) heating excavated soil for volatilization, (3) melting and cooling soils to immobilize contaminant	up to 99%	[6,109,122]
Ultrasound remediation	Soil, Sediment, Water	High ultrasonic sound (150–2000 kHz) leading to desorption produced by local turbulence and/or to degradation due to free radical oxidation reactions	~5%, usually used with bioremediation	[123]

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
