# Peer review of "Mercury (Hg) Contaminated Sites in Kazakhstan: Review of Current Cases and Site Remediation Responses"

_ijerph, 2020, doi:10.3390/ijerph17238936_

Round 1

Reviewer 1 Report

The authors of the text set themselves the task of gathering the current research on mercury pollution in two regions in Kazakhstan.
The summary of the knowledge on this subject is valuable as it also includes research published in Russian, which is not always easily accessible to researchers from the West.

However, things that, in my opinion, need to be changed / supplemented for the work to meet the requirements set by the editor of the IJERPH journal:

  • it is possible to find the values ​​of the maximum limit concentration ​​of mercury in various components of the environment also in other EU countries than the Netherlands, especially in the specified "working environment". In the absence of such regulations, guidelines for local authorities and inspectorates can be found. "Not found" sounds a little unprofessional.
  • Search methodology is described in too much detail. There is no need to put the way of searching of the publications.  
  • I do not see any point in including Table 5 as it stands. Remediation methods are listed there, and this is not the purpose of the publication. The list of remediation methods can be found in dedicated books or publications devoted specifically to this issue. This table only makes sense for listing and discussing the remediation methods used in these particular cases in Kazakhstan.
  • Chapter 6 is actually a repetition of information from chapter 4 and table 2, as well as a discussion of table 4. In my opinion, the structure of these chapters and the sense of discussing and in some places duplicating information should be reconsidered.
  • In the conclusions and discussions, I suggest including an evaluation of the method of conducting the presented research, evaluation of conclusions from these publications, whether they are consistent or divergent, showing their possible weaknesses, etc. Thanks to this, the presented manuscript will be more than just a list of publications on a given topic, but also an attempt to evaluate them and setting directions for further research. This will significantly increase the scientific value of the presented work.

Reviewer 2 Report

1)- Provide a good conceptual scheme of the review done, and define the scope of the review clearly in the form of a graphical abstract.
2)- Ensure that the review should provide broad perspective and interest to the readers from different disciplines and countries.

3)This work is mainly from Kazakhstan. Please justify why this work should publish in international Journal and not in Local one.
4)- Do not merely summarize all the existing literatures in different paragraphs (facilitate discussion of key concepts that are relevant for practical applications in research and practice). In each and every paragraph, there has to be at least 5-8 recent references. Many references are very OLD. Include some recent refernces on Hg: such as a) Mercury remediation potential of Brassica juncea (L.) Czern. for clean-up of flyash contaminated sites.

b) Brassica juncea (L.) Czern. (Indian mustard): a putative plant species to facilitate the phytoremediation of mercury contaminated soils
5) - Provide critical scientific discussion to support new/emerging concepts in the field.

6) - Provide more ORIGINAL infographics (minimum 2 nice original figures) that describes the concepts/technologies or mechanisms that was reviewed.
7) - Provide industrial case-studies not only from Kazakhistan but also its comparison with the world (In detail). In this paper, that aspect is limited.
8) - Review of existing patents is also highly encouraged. Example: kindly refer Asian, EU and American patent databases. Please add one new section on this topic.
9) - The practical relevance of your review should be integrated in the discussion so that the global audience receives sufficient information on the practical implication of this review.

10)- Conclusions should be streamlined with the main objectives of this review.

Reviewer 3 Report

In general, the structure of the article is good, however, ideas should be better organized particularly in the Introduction. There
is the intensive use of abbreviations, especially when they are used only once or twice in the manuscript. Methylmercury is not exhaustively reviewed and therefore keywords should be carefully revised. Despite the manuscript is well written, some sentences are hard to read and interpret.

Reviewer 4 Report

General comments:

The manuscript deals with "mercury contaminated sites".

These contaminants include heavy metals that can endanger public health if incorporated in the food chain. Heavy metals are present in different types of industrial effluents and are responsible for environmental pollution.

1. Do not divide the introduction into subheadings.

Merge 1.1, 1.2 and 1.3 together

2. Page 5, Line 146 "2. Effect of Site-specific..." It is better the mechanism is shown in a graph.

3. Page 11, Line 197; "3.1. History" It is better that authors make this part shorter and merge it with "1. Introduction".

4. Page 14, Line 340, "4.1. History" Same comment as above.

5. Page 18, Table 5; You need to mention the removal values of each method.

In my opinion, this paper is more suitable for a book rather than a scientific journal.

Lack of the discussion about remediation methods, removal values, and advantages of the removal methods.

Reviewer 5 Report

See attached file

Reviewer 6 Report

This manuscript summarizes and discusses contamination of Hg of Kazakhstan's middle and northern regions, and related remediation technologies. Introduction needs references. Regulations, search methodology and the relationship site conditions-Hg mobility are well detailed, as well as the remediation response. Conclusions are clear. Few sections need to be slightly re-arranged, including the abstract.

Minor revision is recommended. Below specific comments

Abstract is dispersive. If possible, organize it following the structure of the manuscript (i.e.: making clear aim, methodology and related results).

Line 38. Revise grammar

50: ‘utilizing Hg in their products’ is not necessary

48-51: this sentence needs references: https://doi.org/10.1016/j.jclepro.2019.03.335 with regard to tannery industry, and other refs to address other significant sectors

66-67, 70-71: these sentences need refs

148 ‘with soils’, remove ‘the’

261-264: such conclusions need to be explained, not just mentioned

3.8: the title has to be more specific, i.e.: risk assessment related to who/what

All the paragraphs of section 3 and 4, in which pollution in soil, water, etc. is presented, would beneficiate of a mention of/ comparison with the related limits presented in the section addressing regulations. Otherwise, without a comparison, numbers are just numbers.

Please, make clear in the introduction of Section 5 that references presented in Table 5 are/are not studies related to Hg remediation specifically in the area of interest

The title ‘Further discussion’ is not appropriate. Please, incorporate this section where appropriate or choose a title which is more explanatory.

Round 2

Reviewer 1 Report

The work has been improved in a number of places so that its scientific quality has increased. My comments were taken into account. I still do not see the point of including Table 5, but it does not diminish the quality of the work. If the Authors and other Reviewers see the sense of including this table, I will not object. One can simply apply the Latin sentence here: repetitio est mater studiorum.

Reviewer 2 Report

The revision has significantly improved the content.  I think the content is ready for publication after the approval of other reviewers.

Cheers

Reviewer 3 Report

The manuscript “Mercury (Hg) Contaminated Sites in Kazakhstan: Review of Current Cases and Site Remediation Responses” was revised by the authors and has improved considerably. However, I still find it hard to read due to the use of long sentences and some repetition of ideas. Some parts of the manuscript are restricted to textbook theory, which does not add information relevant to the goal of the article itself. Accordingly, this review is quite long and its extension could be reduced a little. 

Reviewer 4 Report

Reviewers' comments have been addressed.
